# Evaluation of a Lipopolysaccharide and Resiquimod Combination as an Adjuvant with Inactivated Newcastle Disease Virus Vaccine in Chickens

**DOI:** 10.3390/vaccines10060894

**Published:** 2022-06-03

**Authors:** Bal Krishnan Sharma, Saravanan Ramakrishan, Abinaya Kaliappan, Mithilesh Singh, Ajay Kumar, Satyabrata Dandapat, Sohini Dey, Madhan Mohan Chellappa

**Affiliations:** 1Immunology Section, Indian Veterinary Research Institute, Bareilly 243122, India; balkrishansha@gmail.com (B.K.S.); drabinaya1994@gmail.com (A.K.); mithilesh.singh@icar.gov.in (M.S.); satyabrata.dandapat@icar.gov.in (S.D.); 2Division of Animal Biochemistry, Indian Veterinary Research Institute, Bareilly 243122, India; ajay.kumar2@icar.gov.in; 3Division of Veterinary Biotechnology, Indian Veterinary Research Institute, Bareilly 243122, India; sohini.dey@icar.gov.in (S.D.); c.mohan@icar.gov.in (M.M.C.)

**Keywords:** TLR4, TLR7, resiquimod, LPS, adjuvant, combination, NDV

## Abstract

Various toll-like receptor (TLR) agonists have shown potential as adjuvants with different vaccines in both human and livestock species, including chickens. Our previous studies on combination of lipopolysaccharide (LPS; TLR4 agonist) and resiquimod (R-848; TLR7 agonist) showed the synergistic up-regulation of pro-inflammatory Th1 and Th2 cytokines in chicken peripheral blood mononuclear cells (PMBCs). Hence, the present study aimed to explore the combined adjuvant effect of LPS and R-848 with inactivated Newcastle disease virus (NDV) vaccine in chickens. Two weeks-old SPF chickens were immunized with inactivated NDV vaccine along with a combination of LPS and R-848 as an adjuvant with suitable control groups. A booster dose was given two weeks later. Antibody responses were assessed by enzyme linked immunosorbent assay (ELISA) and hemagglutination inhibition (HI) test, while cell-mediated immune responses were analyzed by a lymphocyte transformation test (LTT) and flow cytometry following vaccination. Two weeks post-booster, the birds were challenged with a velogenic strain of NDV, and protection against clinical signs, mortality and virus shedding was analyzed. The results indicated that inactivated NDV vaccine with R-848 induced significantly higher humoral and cellular immune responses with 100% protection against mortality and viral shedding following a virulent NDV challenge. However, the combination of LPS and R-848 along with inactivated NDV vaccine produced poor humoral and cellular immune responses and could not afford protection against challenge infection and virus shedding when compared to the vaccine-alone group, indicating the deleterious effects of the combination on antigen-specific immune responses. In conclusion, the combination of LPS and R-848 showed the inhibitory effects on antigen-specific humoral, cellular and protective immune responses when used as an adjuvant with inactivated NDV vaccines in chickens. This inhibitory effect might have occurred due to systemic cytokine storm. A nanoparticle-based delivery of the combination of LPS and R-848 for slow and sustained release could be tried as an alternative method to explore the synergistic effect of the combination as an adjuvant in chickens.

## 1. Introduction

Various molecular adjuvants such as Toll-like receptor (TLR) agonists have shown potential adjuvant capabilities in both mammals and birds. TLRs recognize and bind with conserved molecular structures of the pathogens, called pathogen-associated molecular patterns (PAMPs), leading to the activation and production of proinflammatory cytokines and up-regulation of MHC and co-stimulatory molecules [1]. These responses interlink TLR-mediated innate responses with adaptive immune response [2,3]. To date, 10 TLRs were reported in chickens, which consist of cell-surface located TLR1a, TLR1b, TLR2a, TLR2b, TLR4, TLR5 and TLR15 and intracellular endosome-located TLR3, TLR7 and TLR21 [4]. 

Lipopolysaccharide (LPS) is the most widely known immunomodulatory molecule, which is recognized by TLR4 [5,6]. In humans, monophosphoryl lipid A (MPLA), a LPS derivative has been approved as an adjuvant in hepatitis B vaccine, fendrix™ [7], and papilloma-induced cervical cancer vaccine, cervarix™ [8]. Intra-nasal co-administration of LPS in liposomes encapsulated with inactivated Newcastle disease virus (NDV) in specific pathogen free (SPF) chicken enhanced humoral responses and protection against NDV infection more efficiently than free, inactivated, intra-nasal NDV vaccines [9]. TLR7 recognizes single-stranded (ss) RNA containing poly-U or GU-rich sequences. In addition, the synthetic molecules such as imidazoquinolines including imiquimod (R-837), resiquimod (R-848) and guanosine analogs such as loxoribine can act as a TLR7 agonist [10,11,12]. We reported the adjuvant potential of resiquimod with inactivated NDV vaccine in SPF chicken, and it enhanced humoral and cellular immune responses with complete protection against virulent NDV challenge and virus shedding [13].

The simultaneous use of two or more TLR agonists as in natural infections can lead to cross-talk [14,15] and results in the suppression or synergism of a particular immune response, which could be used wisely to achieve a maximum level of potency with the least safety issues. Recently, we reported the synergistic interaction of a combination of resiquimod (R-848) and LPS on the expression of IFN-β, IFN-γ, IL-4 and IL-1β transcripts in the chicken PBMCs in vitro, indicating the capacity to induce both Th1 and Th2 types of immune responses [16]. The interaction of TLR4 and TLR7 has also been reported in human and mice models. TLR4 and TLR7/8 agonist combination synergistically upregulated IL-12 and IFN-γ in human PBMCs [17] and enhanced the secretion of inflammatory mediators and the expression of surface markers on human dendritic cells [18]. Furthermore, in neonates, the TLR4 and TLR7/8 agonists’ combination synergistically activated cord blood antigen-presenting cells to express high levels of IL-12p70 and IFN-γ [19]. In mice, the combination of TLR4 and TLR7 agonists showed adjuvant potentials with recombinant influenza virus hemagglutinin by stimulating balanced Th1 and Th2 immune responses against viral hemagglutinin and inducing a protective immune response against virulent challenge of influenza virus [20]. Hence, we were intrigued to perform the current study by using the combination of LPS and R-848 as an adjuvant with inactivated NDV vaccines in chicken.

The inactivated NDV vaccine was selected as a model because of its economic importance, worldwide distribution and highly infectious nature of the causative virus, which causes huge mortality. ND is caused by virulent strains belonging to the genus *orthoavulavirus* and species *Avian orthoavulavirus1* (AOAV 1) (commonly referred as Newcastle disease virus) [21]. Live lentogenic, mesogenic strains and inactivated NDV are used globally for the control of ND. Inactivated NDV vaccines are comparatively safe and adjuvants would be useful for enhancing immunogenicity, which can improve immune responses and protection against NDV. 

## 2. Materials and Methods

### 2.1. Experimental Birds and TLR Agonists

SPF chicks (*n* = 82) were provided with ad libitum feed and water and maintained following standard management practices. The total experiment was approved by the Institutional Animal Ethical Committee (IAEC, ICAR-IVRI, File No. F.1-53/2012-13/JD(R) dated 22 May 2015). TLR7 (resiquimod; R-848) and TLR4 agonists (lipopolysaccharide from *E. coli* O26:B6) were procured from InvivoGen, San Diego, CA, USA, and Sigma-Aldrich, St. Louis, MO, USA, respectively.

### 2.2. Bulk Production and Titration of NDV

Virulent Newcastle disease virus (NDV) available in the Avian Immunology Laboratory, Immunology Section, ICAR-IVRI, was bulk propagated in 9-day-old embryonated SPF chicken eggs through the intra-allantoic route. The harvested allantoic fluid having virulent NDV was titrated by the Reed and Muench method using SPF embryonated chicken eggs [22] and expressed as ELD_50_ (median embryo lethal dose).

### 2.3. Evaluation of Adjuvant Activity of TLR Agonists’ Combination

#### 2.3.1. Preparation of Inactivated NDV Vaccine

Inactivation of virulent NDV was performed by using 0.5% formaldehyde, incubation was performed at 4 °C for 24 h. Residual infectivity was checked by inoculating into embryonated chicken eggs (9–11 days old).

#### 2.3.2. Experimental Design

A total of 82 two-week-old SPF chicks were randomly allotted to one of the following seven groups (*n* = 10/group except groups F and G where it was 11/group), as presented in Table 1. Vaccine alone or in combination with TLR agonist(s) was administered by intramuscular route keeping PBS injected birds as the unvaccinated control. A booster dose was given on 14 days post-immunization (dpi). Two weeks post-booster, all birds were challenged with a velogenic strain of NDV (10^5^ ELD_50_ per bird) intramuscularly. Clinical signs and mortality were observed daily until 14days post-challenge (dpc). Cloacal swabs (*n* = 6/group or from the surviving birds) were collected from the birds on day 0, 4, 7 and 14 post-challenge and inoculated into 9-day-old embryonated chicken eggs (*n* = 3 eggs/sample) through the intra-allantoic route. Three-days post-inoculation, the allantoic fluid was checked for NDV growth by spot HA using 10% chicken RBCs.

#### 2.3.3. Evaluation of Humoral Immune Response

Blood was collected at weekly intervals after the vaccination (*n* = 6/group) and challenge (*n* = 6/group or from the surviving birds). Serum was separated and stored at −20 °C until use. Serum samples were analyzed by using the HI test using 1% chicken red blood cells (RBCs) according to the OIE recommended protocol [23]. The HI titer was determined as the highest dilution of serum sample that inhibited the NDV agglutination of chicken RBCs. Viral-specific total antibodies against NDV were also measured in the above-collected serum samples at 7, 14, 21 and 28 dpi as well as on 7 and 14 dpc using the commercially available Newcastle disease virus antibody test kit (IDEXX Laboratories, Westbrook, ME, USA).

#### 2.3.4. Evaluation of Cellular Immune Response

##### Lymphocyte Transformation Test (LTT)

Blood samples (*n* = 6/group) were collected from the jugular vein with anticoagulant at 7, 14, 21 and 28 dpi. The PBMCs were isolated as previously described [16] and analyzed for antigen-specific cellular proliferations at each time point. The cells were suspended in RPMI 1640 complete medium containing 10% foetal calf serum and 100 IU/mL penicillin and 100 µg/mL streptomycin. Cell viability was determined by the trypan blue dye exclusion method, and the concentration was adjusted to 1 × 10^7^ cells/mL. The cells (100 µL) were plated in 96-well sterile tissue culture plates. RPMI 1640 medium (100 µL) with or without ConA (10 µg/mL), NDV (10 µg/well) was added to the wells in triplicate. The plates were incubated at 37 °C with 5% CO_2_ for 72 h in a humidified chamber. At the end of incubation, 20 µL of MTT [3-(4,5-dimethylthiazol-2-yl-2,5-diphenyl-tetrazolium bromide; Sigma-Aldrich, St. Louis, MO, USA] was added from the stock (5 mg/mL). The plates were re-incubated at the same condition for 4 h. Then, 100 µL of the culture supernatant was discarded from each well. The formazan crystals formed were dissolved by the addition of dimethyl sulfoxide (Amresco, Solon, OH, USA), 100 µL to each well, and a reading was taken at A_570_ optical density (OD) on a microplate ELISA reader (Bio-Rad, Hercules, CA, USA). The blastogenic response was calculated by dividing the mean OD of the stimulated cultures by the mean OD of unstimulated control [24] and expressed as the mean stimulation index (SI). 

##### Flow Cytometry Analysis

The PBMCs (*n* = 6/group) isolated from the experimental SPF chicks were analyzed at two weeks after primary vaccination and one week after booster vaccination by flow cytometry for CD4+ and CD8+ T cells. For analysis, 2 × 10^5^ cells were incubated with R-phycoerythrin (RPE) labeled mouse anti-chicken CD4+: RPE or CD+8a: RPE monoclonal antibodies (AbDSerotec Ltd., Oxford, UK) for 30 min in the dark. Subsequently, the cells were washed twice with PBS containing 2% fetal bovine serum (FBS), and aliquots of 10,000 cells were analyzed per sample by BD FACS calibur flow cytometer. Unstained cells were included as a negative control.

### 2.4. Challenge Study

Two weeks after the booster dose, the experimental birds were challenged with 10^5^ ELD_50_ of virulent NDV through the intramuscular route. The birds were closely monitored for the clinical signs and mortality up to day 14 post-challenge.

### 2.5. Viral Shedding

Cloacal swabs (*n* = 6/group or from the surviving birds) were collected from the experimental birds in 50% glycerol-PBS containing antibiotics at 0, 4, 7 and 14 days after challenge for the assessment of viral shedding. The samples were vortexed, and the supernatant was collected after centrifugation at 1000× *g* for 15 min at 4 °C. After filtration through a 0.45μm syringe filter, the samples were inoculated into 9-day-old embryonated chicken eggs through the intra-allantoic route. Three eggs per sample were used. Eggs were incubated at 37 °C for 3 days and were candled every 24 h to check for the death of any embryo. Embryos that died at 24 h were discarded due to the non-specific cause of death. After 72 h post-inoculation, all eggs were chilled at 4 °C for 12 h and checked for NDV growth by using the spot hemagglutination test using 10% chicken RBC.

### 2.6. Statistical Analysis

The treatment effect at each point of time was assessed by two-way analysis of variance (ANOVA) with Tukey’s test as the post hoc test to find the significance of pair-wise mean difference. The minimum level of significance was set at 95%, and results were presented as Mean ± SE. The survival curve was plotted according to the Kaplan–Meier method and the significant difference was determined using the log-rank (Mantel–Cox) test. The 2 × 2 contingency table on viral shedding post-challenge was analyzed by the Chi-square test corrected for Yate’s correction as some cell frequencies were < 5. Mean differences were considered significant when *p* < 0.05. GraphPad Prism 8.0.1 (Graphpad Software Inc., San Diego, CA, USA) was used for statistical analysis.

## 3. Results

### 3.1. Preparation of Inactivated NDV Vaccine

The bulk production of NDV was conducted in SPF chicken embryonated eggs. The virulent NDV caused the death of embryos within 48h of inoculation and produced lesions, such as whole-body congestion and hemorrhagic changes. The ELD_50_ of the bulk propagated virulent NDV was 5 × 10^8.5^ /mL. Formalin-inactivated NDV neither caused the death of embryos nor any lesions in embryonated eggs, which confirmed the complete inactivation of virulent NDV.

### 3.2. Humoral Immune Response

The humoral immune response was assessed by HI tests on serum samples collected at weekly intervals after vaccination and also during the first and second weeks after challenge. In each group, the mean of log_2_ HI titer was calculated (Figure 1). The HI titer was below the limit of detection in the control birds. At every time point studied, the birds of group E had significantly higher HI titer than other groups except at 14 dpc, where it was in group F (*p* < 0.05). The HI titer was lower in both groups F and G than group C at 14 and 21 dpi. However, the difference was not statistically significant (*p* > 0.05) and group G showed numerically lesser HI titer than that of group F at 14, 21, 28 dpi and 7 dpc. After the virus challenge, a small reduction in HI titer was observed in all groups at 7 dpc than that of 28 dpi and then increases in the same manner at 14 dpc. At 7 dpc, a significant peak HI titer was recorded in group E (*p* < 0.01). All control birds died within 7 days after challenge. 

The adjuvant effect of TLR agonists combination with inactivated NDV vaccine on the NDV-specific antibody was assessed by ELISA using a commercial kit (Figure 2). The specific antibodies to NDV were below the detectable limit in group A after vaccination. The antibody titer was higher in vaccinated groups than that of the control (*p* < 0.05). The group E birds had significantly higher antibody titer than that of other groups at all the time points studied (*p* < 0.05). The group G birds showed lesser antibody titer than group F at 21 dpi. After the challenge infection, a small reduction in antibody titer was observed at 7 dpc in all groups than that of 28 dpi and then it increased in the same manner at 14 dpc. The highest antibody titer was observed in group E at all time points studied. 

### 3.3. Cellular Immune Response

Cellular immune responses were evaluated by LTT and flow cytometry. Lymphocyte proliferation was measured by the MTT colorimetric assay. NDV antigen-specific lymphocyte proliferation was expressed as a stimulation index (S.I) and is depicted in Figure 3. All vaccinated birds showed significantly higher SI than that of the control at 14 and 28 dpi, with the exception of group F and G birds (*p* < 0.05). NDV antigen-specific lymphocyte proliferation was at the maximum in group E than that observed in the other groups (*p* < 0.05). The SI of group F and G was significantly lower than that observed in the control group (*p* < 0.05) at 7, 14 and 21 dpi. Overall, group F and G birds did not show lymphocyte proliferation with ConA in comparison to all other groups (data not shown). 

The analysis of CD4^+^ and CD8^+^ T cells in chicken PBMC was performed by flow cytometry at 14 and 21 dpi. The percentages of CD4^+^ and CD8^+^ T cells were significantly higher in group E than that of the other groups (*p* < 0.05). At 14 dpi, the percentages of CD4^+^ and CD8^+^ T cells in group E were 27.87 ± 0.58 and 11.71 ± 0.59, respectively, and the same at 21 dpi were 30.13 ± 0.62 and 13.24 ± 0.23, respectively. The percentages of CD8^+^ T cells were significantly lower in both groups F and G than that of the control at both 14 and 21 dpi (*p* < 0.05) (Figure 4). 

### 3.4. Protection against Clinical Signs, Mortality and Virus Shedding

About 24 h post-challenge, all control birds showed clinical signs, viz., depression, distress, ruffled feathers, anorexia, greenish diarrhea, trembling and dehydration. Furthermore, all birds of control group died within day 7 post-challenge and post-mortem examination of dead birds showed characteristic lesions of NDV, viz., petechiae in the proventriculus and hemorrhages in the intestine and trachea. The percent mortality rates in groups B, C, D, F and G were 20, 30, 40, 81.82 and 100, respectively. In contrast, there was no mortality even after 14 dpc in group E birds (Table A1). The group F and G birds showed only a delay in death time. The survival curve was statistically significant (*p* < 0.0001) with a chi-square value of 73.13 and df 6 (Figure 5).

Virus shedding after the challenge was also measured. No NDV shedding was recorded in group E birds at any time interval, and all other group birds showed virus shedding at all intervals (Table 2). Group E showed a significant difference in virus shedding when compared to groups B or C on day 4 and 7 post-challenge (*p* = 0.0039). Group E showed a higher protection against viral shedding than vaccine alone group (95% CI: 2.89–9885) at day 4 and 7 post-challenge.

## 4. Discussion

Based on the immunostimulatory properties of TLR agonists, many attempts have been made to employ them as an adjuvant with various vaccine antigens. In chicken, TLR agonists have been used as vaccine adjuvants as well as prophylactic agents [25,26]. Emerging evidence indicates that the involvement of multiple TLRs with their respective ligand could result in cross-talk between them. Several studies demonstrated that the activation of two or more TLRs simultaneously can cause interactions with each other and produce either a synergistic or antagonistic response [27,28,29,30]. In mice, TLR4 and TLR7 worked additively to induce antigen-specific IgG and provided protection against influenza challenges when this combination was used as an adjuvant [31]. Recently, we reported that a combination of R-848 and LPS synergistically up-regulated the expression of IFN-β, IFN-γ, IL-4 and IL-1Β in the chicken PBMC in vitro [16]. In the present study, the adjuvant potential of the combination of R-848 and LPS with an inactivated NDV vaccine was evaluated.

Humoral immunity plays a crucial role in host defense against the invading microorganism. Antibodies contribute to virus neutralization by complement activation or opsonization by NK cells, macrophages and monocytes, thus indicating the importance of antibody titers in resistance to NDV infection. NDV-specific virus neutralization antibody is considered as the best correlate of the immune status of birds and, thus, for protection [32].

In the present study, both HI and ELISA titers were significantly higher in birds receiving inactivated ND vaccine with R-848 than that of other groups. This is supported by our earlier report, wherein R-848 enhanced antibody titers when used along with inactivated NDV vaccine in SPF chicken [13]. In contrast, the combination of LPS and R-848 with inactivated vaccines induced lesser antibody titers than that of the vaccine alone. Furthermore, the combination in low dose (25 µg each) showed higher antibody titersthan that of high dose (50 µg each), along with an inactivated vaccine indicating the inhibitory effect of a higher dose combination on humoral immune responses. This was in agreement with a previous study [33] wherein a higher dose combination failed to improve the antibody response against *Plasmodium falciparum* cirumsporozoite vaccine in mice.

In addition to antibodies, cell-mediated immunity (CMI) also plays an important role in protection and clearance of NDV infection from the body [32,34]. Cell-mediated immunity is considered as a major protective mechanism against viral diseases [35]. The antibody alone is the key modulator for protection but the intracellular elimination of NDV infected cells depends upon CMI, and this also helps in reducing viral shedding from the body [36]. 

In the current study, we found that the combination of LPS and R-848 along with inactivated NDV vaccine showed inhibitory effects on cellular immune responses, as indicated by significantly lower SI in LTT as well as CD8+ T cells in flow cytometry analysis in both groups F and G (combination of LPS and R-848 along with inactivated NDV vaccine) than that of the control. The combination of LPS and R-848 with inactivated NDV vaccine groups (group F and G) showed an SI of less than one with ConA stimulation (data not shown), which indicates the immunosuppressive effect of the combination. Furthermore, the inhibitory effect on cellular immune response increased with increasing doses of the agonists’ combination. However, the combination of TLR4 and TLR7 agonists along with antigen-induced MHC class I and class II restricted T cell responses in mice [20]. This might be due to the species difference between mice and chicken.

It should be noted that the co-stimulation of TLR4 and TLR7/8 with their respective ligands has shown different outcomes in various animal models due to the differential expression pattern of these receptors. TLR4 and TLR7/8 agonist combinations have shown a lack of synergism against various pathogens in animals [37,38,39]. The combination of TLR4 and TLR7/8 agonists failed to enhance humoral and cellular immune responses when used with the ID83 fusion protein of *Mycobacterium bovis* in cattle [37]. In mini pigs, the combination of TLR4 and TLR7/8 agonists exhibited an additive effect on antigen-specific immune responses through the intradermal route while the same combination produced no effects through the intranasal route [38]. In another study, the co-administration of TLR4 and TLR7/8 did not show synergism when used with soluble *Leishmania* antigen in mice [39].

In our study, the combination of LPS and R-848 could not induce more antibody and cellular responses along with inactivated NDV vaccine compared to the vaccine alone group. The combination of LPS and R-848 induces a synergistic up-regulation of IL-1β, IFN-β, IFN-γ and IL-4 transcripts in the chicken’s immune cells, as reported earlier [16], and could lead to the production of more cytokines (cytokine storm). The cytokine storm might be the possible reason for the deleterious effects of the combination of LPS and R-848 on antigen-specific antibodies and cellular immune responses when used with inactivated NDV vaccines in chickens. Systemic cytokine production can be reduced by the encapsulation of TLR agonists into a nanoparticle [40]. This is supported by various reports in which the co-localization of TLR4 and TLR7 agonists in a nanoliposome enhanced cytokine response and eliminated the deleterious effects [41]. Furthermore, the nanoparticle delivery of R-848 led to a strong induction of immune responses and less toxicity and cytokine production than compared to free R-848 [42]. Several studies also underpin that the nano-delivery of TLR4 and TLR7 could enhance their synergism and reduce reactogenicity [43,44,45].

In addition, this study also revealed that the combination of LPS and R-848 in low doses induced higher humoral as well as cellular immune responses compared to the same combination in high doses. The reason behind this may be the fact that a higher dose of the agonist’s combination causes increased cytokine storms, resulting in deleterious effects compared to low doses. Low doses (less than 25 µg, each) of the agonist combination or slow release such as nanoparticle delivery of this combination with inactivated vaccine might be a good choice in the future.

The efficacy of the vaccine regimen on the induction of protections in the immune response against NDV was evaluated by a challenge study. Antibodies to the HN and F glycoprotein are critical for virus neutralization and, thus, protection from a vNDV challenge [32]. These antibodies block the attachment and fusion of virus to the host cell and provide protection against viral infection in chickens, which is directly correlated with the serum HI titer [46]. In the present study, the inactivated NDV vaccine alone group showed 70% protection while it was 80% with commercial vaccines. LPS along with inactivated NDV vaccine (Group D) protected only 60 % of the birds in the challenge. Though the combination of LPS and R-848 along with the inactivated NDV vaccine delayed the death, it could not induce protections against virulent NDV challenges. Furthermore, the combination in low dose showed more protection than that of high doses along with inactivated NDV vaccine. In contrast to this, the combination of TLR4 and TLR7 agonists reduced mortality and morbidity in mice against virulent challenges of influenza when it was used as an adjuvant with influenza antigens [20]. This might be due to the species difference between mice and chicken.

The amount of NDV shed into the environment by vaccinated birds is a potential indicator of vaccine efficacy [47,48]. In the present study, we have evaluated virus shedding from each vaccinated group after challenges with virulent NDV. Our results indicated that the inactivated vaccine alone groups and the vaccine plus the combination of LPS and R-848 could not protect the birds against virus shedding after virulent NDV challenges, which might be due to the poor cell-mediated immune responses as the elimination of intracellular pathogens is critically dependent on CD8^+^ T cells.

## 5. Conclusions

In summary, the combination of LPS and R-848 with an inactivated NDV vaccine caused inhibitory effects on humoral as well as cellular immune responses, and it also could not protect against mortality and virus shedding after virulent NDV challenges. The deleterious effect of the combination of LPS and R-848 might be due to the high amount of systemic cytokine production. Low doses of the agonists or nanoparticle delivery of LPS and R848 along with inactivated NDV vaccine can be attempted as it can reduce the cytokine storm in vivo and might help in enhancing antigen-specific immune responses.

## Figures and Tables

**Figure 1 vaccines-10-00894-f001:**
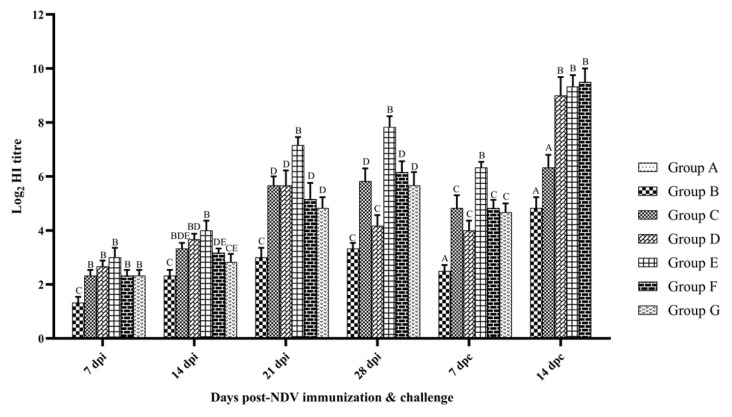
Hemagglutination inhibition antibody titers in SPF chicken following immunization and challenge. Treatment effect was analyzed at each time point by two-way ANOVA with Tukey’s test as post hoc test. The probability of an alpha error was set at 0.05. Different superscripts above bars (mean ± SE) indicate significant difference between groups (*p* < 0.05). dpi: days post-NDV immunization; dpc: days post-challenge. Group A: Control; group B: Commercial NDV vaccine; group C: inactivated NDV vaccine; group D: inactivated NDV vaccine with LPS; group E: inactivated NDV vaccine with R-848; group F: inactivated NDV vaccine with LPS (25 µg) + R-848 (25 µg) and group G: inactivated NDV vaccine with LPS (50 µg) + R-848 (50 µg).

**Figure 2 vaccines-10-00894-f002:**
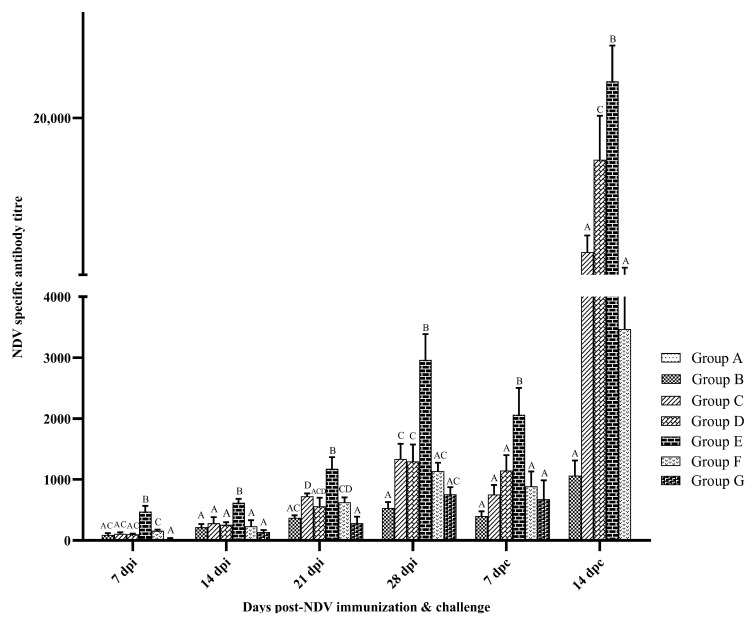
NDV-specific antibody titers in SPF chicken following immunization and challenge. Treatment effect was analyzed at each time point by two-way ANOVA with Tukey’s test as the post hoc test. The probability of an alpha error was set at 0.05. Different superscripts above bars (mean ± SE) indicate significant difference between groups (*p* < 0.05). dpi: days post-NDV immunization. Group A: Control; group B: Commercial NDV vaccine; group C: inactivated NDV vaccine; group D: inactivated NDV vaccine with LPS; group E: inactivated NDV vaccine with R-848; group F: inactivated NDV vaccine with LPS (25 µg) + R-848 (25 µg) and group G: inactivated NDV vaccine with LPS (50 µg) + R-848 (50 µg).

**Figure 3 vaccines-10-00894-f003:**
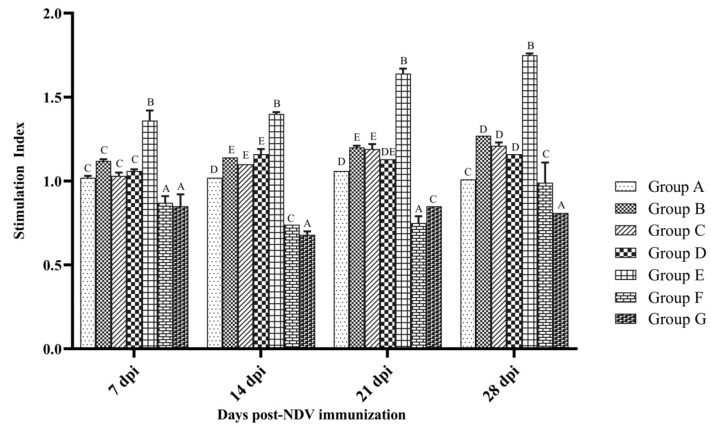
Lymphocyte transformation specific to NDV antigen in PBMCs collected from SPF chicken following immunization. Treatment effect was analyzed at each time point by two-way ANOVA with Tukey’s test as post hoc test. The probability of an alpha error was set at 0.05. Different superscripts above bars (mean ± SE) indicate significant difference between groups (*p* < 0.05). Group A: control; group B: commercial NDV vaccine; group C: inactivated NDV vaccine; group D: inactivated NDV vaccine with LPS; group E: inactivated NDV vaccine with R-848; group F: inactivated NDV vaccine with LPS (25 µg) + R-848 (25 µg) and group G: inactivated NDV vaccine with LPS (50 µg) + R-848 (50 µg).

**Figure 4 vaccines-10-00894-f004:**
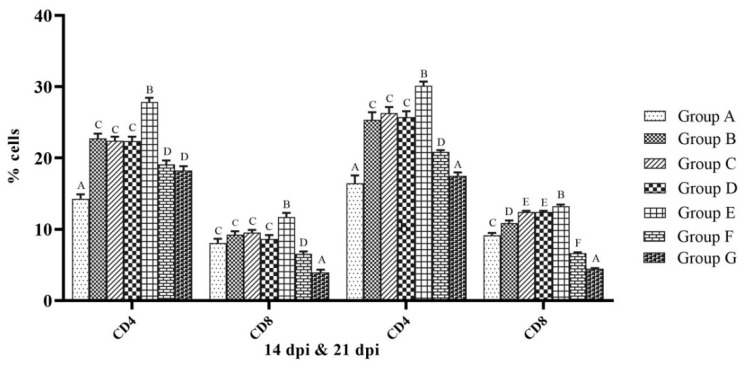
Percentage of CD4^+^ and CD8^+^ T cells in PBMCs collected from SPF chicken analyzed by flow cytometry. Treatment effect was analyzed at each time point by two-way ANOVA with Tukey’s test as post hoc test. The probability of an alpha error was set at 0.05. Different superscripts above bars (mean ± SE) indicate significant difference between groups (*p* < 0.05). dpi: days post-NDV immunization. Group A: control; group B: commercial NDV vaccine; group C: inactivated NDV vaccine; group D: inactivated NDV vaccine with LPS; group E: inactivated NDV vaccine with R-848; group F: inactivated NDV vaccine with LPS (25 µg) + R-848 (25 µg) and group G: inactivated NDV vaccine with LPS (50 µg) + R-848 (50 µg).

**Figure 5 vaccines-10-00894-f005:**
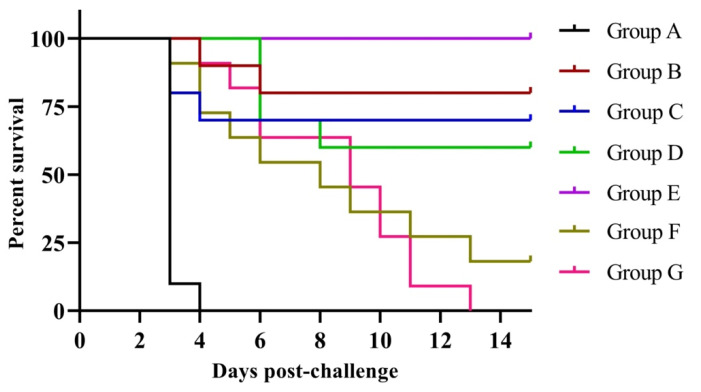
Survival analysis (Kaplan–Meyer analysis) of experimental SPF chicken following challenge with virulent Newcastle disease virus. The survival curve was statistically different (*p* < 0.0001) with a chi-square value of 73.13 and df 6. Group A: control; group B: commercial NDV vaccine; group C: inactivated NDV vaccine; group D: inactivated NDV vaccine with LPS; group E: inactivated NDV vaccine with R-848; group F: inactivated NDV vaccine with LPS (25 µg) + R-848 (25 µg) and group G: inactivated NDV vaccine with LPS (50 µg) + R-848 (50 µg).

**Table 1 vaccines-10-00894-t001:** Immunization plan followed in the experimental birds.

Groups (*n*=10 or 11/Group) *	Vaccine (Dose/Bird)	TLR Agonist(s) (Dose/Bird)
Control (A)	PBS	-
Commercial vaccine (B)	Commercial oil adjuvanted inactivated NDV vaccine (recommended dose)	-
Vaccine alone (C)	Inactivated NDV vaccine (≥10^7^ EID_50_)	-
Vaccine + TLR4 agonist (D)	Inactivated NDV vaccine (≥10^7^ EID_50_)	LPS (50 µg)
Vaccine + TLR7 agonist (E)	Inactivated NDV vaccine (≥10^7^ EID_50_)	R-848 (50 µg)
Vaccine + TLR4 and TLR7 agonists in low dose (F)	Inactivated NDV vaccine (≥10^7^ EID_50_)	LPS (25 µg) + R-848 (25 µg)
Vaccine + TLR4 and TLR7 agonists in high dose (G)	Inactivated NDV vaccine (≥10^7^ EID_50_)	LPS (50 µg) + R-848 (50 µg)

* The number of birds was 10 in each group except groups F and G, where it was 11/group. +, indicate usage in combination.

**Table 2 vaccines-10-00894-t002:** Viral shedding from cloacal swabs of experimental SPF chicken following challenge with virulent Newcastle disease virus.

Group ^$^ (*n* = 6/Group)	Days Post-Challenge		
	4	7	14
A	Died	Died	Died
B	100 (6)	100 (6)	100 (6)
C	100 (6)	100 (6)	66.67 (4)
D	66.67 (4)	66.67 (4)	33.33 (2)
E	0 (0)	0 (0)	0 (0)
F	66.67 (4)	66.67 (4)	33.33 (2) *
G	66.67 (4)	66.67 (4)	Died

Values in the parenthesis indicate the number of positive observations for six birds tested. ^$^ Group A: control; group B: commercial NDV vaccine; group C: inactivated NDV vaccine; group D: inactivated NDV vaccine with LPS; group E: inactivated NDV vaccine with R-848; group F: inactivated NDV vaccine with LPS (25 µg) + R-848 (25 µg) and group G: inactivated NDV vaccine with LPS (50 µg) + R-848 (50 µg). * Only two surviving birds were tested for viral shedding.

## Data Availability

Data will be made available upon reasonable requests.

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
