# Peer review of "Evaluation of a Lipopolysaccharide and Resiquimod Combination as an Adjuvant with Inactivated Newcastle Disease Virus Vaccine in Chickens"

_vaccines, 2022, doi:10.3390/vaccines10060894_

Round 1

Reviewer 1 Report

A well-designed study for evaluating the combination of LPS and R848 as an adjuvant with inactivated NDV vaccine in the chicken.

The manuscript is well written with no major concerns. The minor comments are:

1- I think the base of this study was positive in vitro study data in addition to published studies related to viruses other than NDV. I did not see any discussion on correlating in vitro study with in vivo study in the discussion section.

2- Lines 87-88- new nomenclature name should be used. Similarly, reference can be replaced with a recent reference with new nomenclature.

3- Method line 175- not sure one passage for 3 days is good for evaluating virus shedding.

4- Discussion is too long and can be reduced significantly by focusing on correlating the study data. Discussion is more focused on inactivated NDV vaccine with R-848 which has been previously published so it looks like a review. Another example like virus shedding paragraph, no need to explain the mechanism.

Is there any study showing the synergistic effect of LPS and R848 with any virus vaccine for birds or animals? If yes, then it should be included in the discussion.

5- References - mostly old references. I think number of references can be reduced.

Author Response

Thanks a lot for the encouraging comments and critical appraisal of our manuscript

Reviewer 2 Report

This article evaluated the inactive NDV vaccine combined with R-848 (TLR7 agonist) and LPS (TLR4 agonist) as adjuvants in chickens. Not only each adjuvant individual function was discussed, but the combination of two adjuvants and the dose of two adjuvants were explored. Beside these, it also employed various methods to assess the effect after vaccination chickens. Though this work is pretty good, there were also some problems.

1: There were some grammar mistakes and words misuse in writing. For example, Line 85, “study” should be erased from “The inactive NDV vaccine study”. In lines 90-91, “lead to improved” should write into “lead to improve”. The legend of Figure 1-3 “Day post-NDV immunization” should write into “Days post xxx”. Line 333, “enhanced significant higher” is repetition, “enhanced” means “higher”.

2: In this paper, for HA experiment, you used 10% chicken RBCs (lines 122 and 177). While for HI, you used 1% chicken RBCs (129). Please make sure if 10% chicken RBCs was correct.

3: To immunize the experimental chickens, you designed Group F and Group G with low and high doses of adjuvants. Could you please clarity how do you define the threshold of low and high dose?

4: Table 2 and Figure 5 presented the same data, it was a little repetition. Maybe put table 2 in the appendix is better.

Author Response

(The authors gave the same response as above.)

Reviewer 3 Report

The manuscript describes the deleterious effects of the combinational use of LPS and resiquimod as adjuvants for chicken NDV vaccine. All the data are very clear, and properly presented. Some minor modifications pointed out  below would improve the manuscript. 

1. Table 1 or somewhere in M&M: The content (molecular species) of adjuvant used in Commercial vaccine should be described.

2. page 2, line87: "NDV" means Newcastle disease? If so, this should be "ND".

3. Table 2 and page12, line394: The data of B and C may be incorrectly written.

4. page 11, line 375: The word "unwanted" should be changed to "deleterious" or other more proper word.

5. page 12, line 401-404: The authors should describe here the possible reason(s) for the difference of adjuvant effects between mice and chickens.

Author Response

(The authors gave the same response as above.)
